# Chemical Characteristics and Source of PM$_{2.5}$ in Hohhot, a Semi-arid City in Northern China: Insight from the COVID-19 Lockdown

Haijun Zhou[1,2,3#], Tao Liu[4#], Bing Sun[5], Yongli Tian[4], Xingjun Zhou[4], Feng Hao[4], Xi Chun[1,2,3], Zhiqiang Wan[1,2,3], Peng Liu[1], Jingwen Wang[1], Dagula Du[6]

[1]College of Geographical Sciences, Inner Mongolia Normal University, Hohhot 010022, China
[2]Provincial Key Laboratory of Mongolian Plateau's Climate System, Inner Mongolia Normal University, Hohhot 010022, China
[3]Inner Mongolia Repair Engineering Laboratory of Wetland Eco-environment System, Inner Mongolia Normal University, Hohhot 010022, China
[4]Environmental Monitoring Center Station of Inner Mongolia, Hohhot 010011, China
[5]Hohhot Environmental Monitoring Branch Station of Inner Mongolia, Hohhot 010030, China
[6]Environmental Supervision Technical Support Center of Inner Mongolia, Hohhot 010011, China

*Correspondence to*: Haijun Zhou (hjzhou@imnu.edu.cn)

[#] H. Zhou and T. Liu contributed equally to this work.

**Abstract.** A knowledge gap exists concerning how chemical composition and sources respond to implemented policy control measures for aerosols, particularly in a semi-arid region. To address this, a single year's offline measurement was conducted in Hohhot, a semi-arid city in northern China, to reveal the driving factors of severe air pollution in a semi-arid region and assess the impact of the COVID-19 lockdown measures on chemical characteristics and sources of PM$_{2.5}$. Organic matter, mineral dust, sulfate, and nitrate, accounted for 31.5%, 14.2%, 13.4%, and 12.3% of the total PM$_{2.5}$ mass, respectively. Coal combustion, vehicular emissions, crustal sources, and secondary inorganic aerosols were the main sources of PM$_{2.5}$ in Hohhot, at 38.3%, 35.0%, 13.5%, and 11.4%, respectively. Due to the coupling effect of emission reduction and improved atmospheric conditions, the concentration of secondary inorganic components, organic matter, and elemental carbon declined substantially from the pre-lockdown (pre-LD) period to the lockdown (LD) and post-lockdown (post-LD) periods. The source contribution of secondary inorganic aerosols increased (from 21.1 to 37.8%), whereas the contribution of vehicular emissions was reduced (from 35.5% to 4.4%) due to lockdown measures. The rapid generation of secondary inorganic components caused by unfavorable meteorological conditions during lockdown led to serious pollution. This study elucidates the complex relationship between air quality and environmental policy.

# 1 Introduction

With the rapid development of industrialization and urbanization, many developing countries, such as China and India, have suffered severe air pollution, especially from fine particulate matter ($PM_{2.5}$, aerodynamic diameter $\leq 2.5$ μm) (Huang et al., 2019). To improve air quality, China has implemented various clean air policies (Zhang et al., 2019). As a result, the annual mean $PM_{2.5}$ concentration in China decreased from 50 μg/m$^3$ in 2015 (Meec, 2015) to 33 μg/m$^3$ in 2020 (Meec, 2020). However, the mean level of $PM_{2.5}$ is still much higher than the new guideline of the World Health Organization (5μg/m$^3$) (Who, 2021). It is a challenge to decrease the level of $PM_{2.5}$ to such a low level in China, especially in northern China, which consumes the majority of coal for winter heating (Huang et al., 2019; Feng et al., 2022b). Insufficient understanding of the complex relationship between air quality and environmental policy limits the effectiveness of our control measures to improve air quality.

To limit the spread of the COVID-19 pandemic, most cities around the world implemented strict lockdown measures, and the anthropogenic emission of air pollutants was reduced substantially, which in turn has caused considerable changes in the chemical composition and sources of $PM_{2.5}$ (Le et al., 2020; Ma et al., 2021; Chen et al., 2020). The lockdown provided a good opportunity to study the effect of emission reduction on air quality. In addition, this scenario can be used by policymakers to formulate effective policies to prevent atmospheric pollution. Stringent traffic restrictions during the COVID-19 lockdown led to important reductions in the concentrations of elemental carbon, metals, and nitrate in an urban site of the western Mediterranean (Clemente et al., 2022). The substantial reduction in nitrate in the Beijing-Tianjin-Hebei region during the lockdown period was attributed to the drastic reduction in vehicular movement and the suspension of public transport (Sulaymon et al., 2021). Primary pollutants were reported to have decreased dramatically due to the lockdown measures, while secondary pollutants were reported to have increased (Chang et al., 2020; Huang et al., 2021; Zheng et al., 2020). Secondary pollutants like $PM_{2.5}$ and $O_3$ depend more strongly on weather conditions and show a limited response to emission changes in single sectors (Gao et al., 2021a; Matthias et al., 2021).

Extensive studies have been conducted to investigate the responses of atmospheric pollutants to emission reduction during the COVID-19 lockdown measures (Adam et al., 2021; Rodríguez-Urrego and Rodríguez-Urrego, 2020). However, most of the previous studies are based on online observation and/or satellite-derived data and have focused on the changes in atmospheric pollutants, influence of meteorological conditions, and emission reduction (Li et al., 2021; Srivastava, 2021). Relatively few studies have focused on the chemical composition and sources of $PM_{2.5}$ in semi-arid regions, especially using offline measurement. Source apportionment using an online dataset is impeded by the missing information on Si and Al (Gao et al., 2016), resulting in considerable uncertainty in the estimation of dust sources. Mineral dust is considered to be one of the main components of aerosols in semi-arid regions (Kumar and Sarin, 2009; Wang et al., 2016). As a typical semi-arid city of northern China, Hohhot suffers frequent air pollution in spring and winter (Huang et al., 2013b). The chemical characteristics, sources, and their response to implemented control measures in this region are still unclear.

In response to the substantial reduction in anthropogenic emission, the concentration of $PM_{2.5}$ in most of the European cities (Tobás et al., 2020; Collivignarelli et al., 2020; Gualtieri et al., 2020; Gkatzelis et al., 2021; Matthias et al., 2021), Latin American cities (Mendez-Espinosa et al., 2020; Nakada and Urban, 2020; Hernández-Paniagua et al., 2021), US cities (Pata, 2020), Indian cities (Sharma et al., 2020), Chinese cities (Bao and Zhang, 2020), and the southeast Asia region (Kanniah et al., 2020) have decreased substantially, compared to pre-LD periods and/or previous years. However, compared with the decreasing trends of most of the cities in the world, the concentrations of $PM_{2.5}$ in some cities of the North China Plain have increased unexpectedly. An increase ($p<0.01$) in $PM_{2.5}$ was found in Hohhot during the LD period (Figure S1), whereas a considerable improvement was reported in most of the cities globally. The anomalously enhanced nitrate in Tianjin during the LD period is a response to the abnormal increase in relative humidity (Ding et al., 2021). The abnormal increase in $PM_{2.5}$ in northern China during the LD period was probably caused by uninterrupted emissions from power plants and petrochemical facilities, as well as the influence of adverse weather conditions (Gao et al., 2021a). The extreme reduction in anthropogenic emissions did not address the occurrences of severe haze events in northern China because of unfavorable meteorological events (Le et al., 2020; Shi et al., 2021), increased atmospheric oxidizing capacity (Wang et al., 2020), enhanced secondary formation (Chang et al., 2020; Huang et al., 2021), and regional transport (Shen et al., 2021; Lv et al., 2020; Zhang et al., 2021). There is no consensus on the reasons for the unexpected increase in $PM_{2.5}$ during the LD period. It is therefore essential to conduct a comprehensive study on the chemical composition and sources of $PM_{2.5}$ in this region, especially during the LD period.

The main objectives of this study were to (1) identify the chemical characteristics and sources of $PM_{2.5}$ in a semi-arid city, (2) investigate the impact of COVID-19 lockdown measures on the chemical composition and sources, (3) reveal the causes of the rapid increase in $PM_{2.5}$ during different heavy pollution episodes. The results of this study will provide a more comprehensive understanding of $PM_{2.5}$ pollution control in semi-arid regions.

## 2 Material and methods

### 2.1 Study area and sampling

Hohhot (40°51′ – 41°8′ N, 110°46′ – 112°10′ E) is located in the northern part of the North China Plain and the central part of the Inner Mongolia Autonomous Region. It is a core city of the Hohhot-Baotou-Ordos urban agglomeration, with an area of 17,224 km² and 3,496,000 inhabitants (http://www.tjcn.org/tjgb/05nmg/37047.html). Topographically, it is in the alluvial lake basin between the Yinshan Mountains and the Yellow River, with the Daqing Mountains in the north and the Manhan Mountain in the southeast. Hohhot has a typical semi-arid climate, with a mean annual precipitation of 335.2–534.6 mm, which occurs mainly in summer (Xie et al., 2021). Due to the minimal precipitation and dry continental terrain, frequent dust storms occur in spring (Gao et al., 2021b). It has six months of coal-fired heating period (15[th] October–15[th] April the next year) (Xie et al., 2021). The sampling site was

located on the rooftop (approximately 50 m above the ground) of the main building of the Ecological and Environmental Department of the Inner Mongolia autonomous region (Figure 1) and represents a typical semi-arid urban environment. The sampling site is surrounded by residential areas without industrial sources nearby. There is one main road named Tengfei Road 50 m to the east.

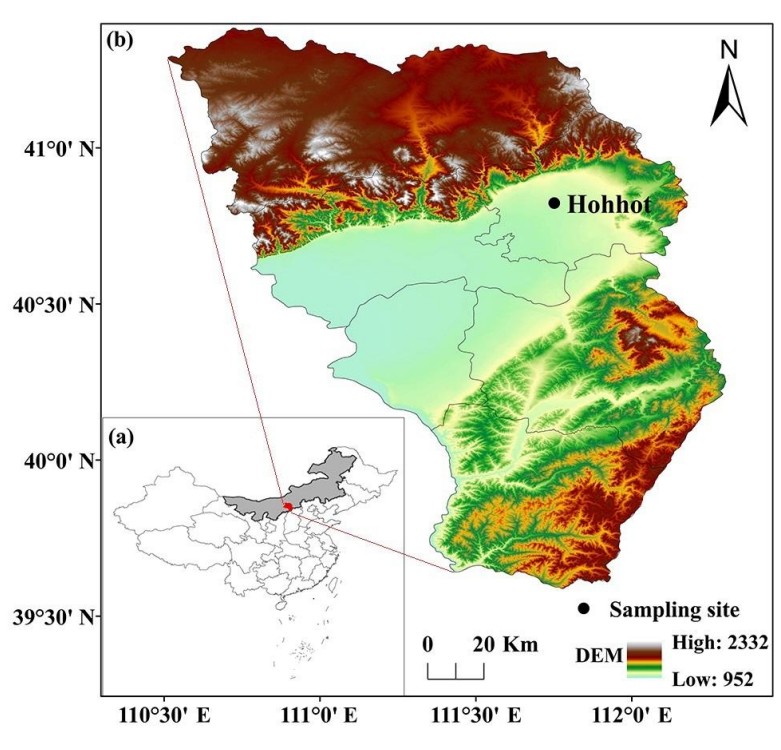

**Figure 1.** Location of (a) Hohhot in China, and (b) the sampling site in Hohhot

The 23-h (10:00 to 09:00 the next day) PM$_{2.5}$ samples were collected in parallel on quartz filters (Pallflex Tissuquartz™, 90 mm, USA) and polypropylene filters (Beijing Safelab Technology Ltd., 90 mm, China) using medium volume air samplers (Model 2050, Qingdao Laoshan Applied Technology Research Institute, China) with a flow rate of 100 L/min. The quartz filters were used for the analysis of water-soluble ions (WSIs), organic carbon (OC), and elemental carbon (EC), while polypropylene filters were used for inorganic elements. A total of 722 PM$_{2.5}$ samples (361 quartz and 361 polypropylene filters) were collected from 8$^{th}$ October, 2019 to 7$^{th}$ October, 2020. Before and after sampling, the quartz filters and polypropylene filters were conditioned for at least 24 h at a stable temperature (20 ± 1 ℃) and relative humidity (50 ± 5%) and then weighed using a microbalance (CP225D, Sartorius, Germany), with a sensitivity of ± 0.01 mg. After weighing, every filter was stored in a preservation box separately at -18 ℃ until analysis. The hourly concentrations of SO$_2$, NO$_2$, CO, and O$_3$ were measured using T100, T200U-NOy, T300, and T400 sensors produced by Automated Precision Inc. (USA), respectively. In addition, the hourly meteorological variables, including relative humidity (RH), wind speed (WS), wind direction (WD), ambient temperature (T), and atmospheric pressure (P) were observed synchronously using an automatic weather station (WS500-UMB, Lufft, Germany).

## 2.2 Chemical analysis

The WSIs (including $SO_4^{2-}$, $NO_3^-$, $NH_4^+$, $Cl^-$, $F^-$, $K^+$, $Ca^{2+}$, $Na^+$, and $Mg^{2+}$) were determined using ion chromatography (Metrohm 881Compact IC Pro, Switzerland). The OCandEC were analyzed using a thermal/optical carbon analyzer (DRI Model 2001, Atmoslytic Inc., USA) following the IMPROVE_A protocol (Chow et al., 2007; Cao et al., 2005). The detailed descriptions of the procedures for WSIs and carbonaceous aerosols can be found in our previous studies (Zhou et al., 2018; Zhou et al., 2016). The inorganic elements, including Si, Al, S, Cl, K, Ca, Ti, V, Cr, Mn, Fe, Cu, Zn, Co, and Pb were analyzed by energy dispersive X-ray fluorescence spectroscopy (Epsilon5, PANalytical B.V., Netherlands) according to the National Environmental Protection standard method of China (HJ 829-2017) and previous studies (Dao et al., 2022; Dao et al., 2021; Chiari et al., 2018). Field blank and replicate analyses were carried out once per 10 samples. The concentrations of field blanks were all lower than the method detection limits, and the relative deviations of replicate analyses were $< \sim 5\%$. All the analytical procedures were strictly controlled according to the referred methods to reduce artificial interference.

## 2.3. Data analysis

The organic matter (OM) and mineral dust (MD) were calculated using equations 1-2 (Xie et al., 2019; Liu et al., 2021). To estimate the secondary formation of inorganic and organic aerosols, the sulfur oxidation ratio (SOR), nitrogen oxidation ratio (NOR), and secondary organic carbon (SOC) were calculated using equations 3-5 (Castro et al., 1999; Fu et al., 2022; Yang et al., 2018):

$$OM = 1.6 \times [OC] \tag{1}$$

$$MD = 2.14 \times [Si] + 1.89 \times [Al] + 1.40 \times [Ca] + 1.43 \times [Fe] + 1.58 [Mn] + 1.21 \times [K] + 1.67 \times [Ti] \tag{2}$$

$$SOR = [SO_4^{2-}] / ([SO_4^{2-}] + [SO_2]) \tag{3}$$

$$NOR = [NO_3^-] / ([NO_3^-] + [NO_2]) \tag{4}$$

$$SOC = OC - EC \times (OC/EC)_{min} \tag{5}$$

## 2.4 Source apportionment

Positive matrix factorization (PMF, version 5.0) was used to estimate source contributions of $PM_{2.5}$ in Hohhot according to the user guide of the United States Environmental Protection Agency (Norris et al., 2014) and a previous study (Paatero and Tapper, 1994). A total of fifteen dominant species ($SO_4^{2-}$, $NO_3^-$, $NH_4^+$, $K^+$, $Na^+$, $Ca^{2+}$, $Mg^{2+}$, OC, EC, Si, Cl, Ti, Fe, Zn, and Pb) were used as input files for the PMF modeling. The displacement (DISP) and bootstrap (BS) methods were conducted to estimate the uncertainty and rotational ambiguity of PMF solutions (Paatero et al., 2014). According to the changes in an output parameter ($Q/Q_{expected}$) and estimation diagnostics analysis (Ulbrich et al., 2009; Liu et al., 2021; Tian et al., 2020), six factors solutions were selected. All of the factors showed a BS mapping above 80 %. The decreased Q values were lower than 0.1 %, and no factor swap occurred. The results indicate that the BS uncertainties can be fully interpreted and the selected solutions were sufficiently robust (Tian et al., 2020; Wang et al., 2021). The summary of error estimation diagnostics from BS and DISP are shown in

 Tables S1–S8. The source profiles of PMF are shown in Figures S2–S9.

## 3 Results and discussion

### 3.1 Temporal variation in PM$_{2.5}$ and chemical composition

Hohhot initiated a Level-I public health emergency response control action on 25[th] January, 2020 (https://www.nmg.gov.cn/zwyw/tpxw/202001/t20200125_256791.html) and downgraded it to a Level-III response on 25[th] February, 2020 (https://www.nmg.gov.cn/zwyw/tzgg/202002/t20200225_258821.html). During this period, complete lockdown measures were taken to prevent the transmission of the SARS-CoV-2. In order to estimate the impacts of lockdown measures on the air quality, we compared the atmospheric pollutants during pre-LD period (25[th] December, 2019 to 24[th] January, 2020), LD period (25[th] January, 2020 to 24[th] February, 2020), and post-LD period (25[th] February, 2020 to 24[th] March, 2020). The daily concentration of PM$_{2.5}$ varied from 4.0 to 293.8 μg/m$^3$, with an annual mean concentration ($\pm$ standard deviation) of 42.6 $\pm$ 40.2 μg/m$^3$, which is higher than the annual mean concentration limits (35 μg/m$^3$) of the National Ambient Air Quality Standards (NAAQS, GB 3095-2012). There were 51 daily PM$_{2.5}$ concentrations higher than the 24-h average concentration limit (75 μg/m$^3$) of NAAQS, accounting for 14.1% of the total number of sampling days. Furthermore, most of them occurred in the heating period, particularly with a predominant wind direction from the southeast. The high intensity of coal combustion for heating discharges a large number of gaseous pollutants (SO$_2$, NO$_2$, and CO), coupled with unfavorable meteorological conditions (high RH and low WS; Figure 2a, 2b), lead to the rapid accumulation of air pollutants (Figure 2c, 2d, and 2e).

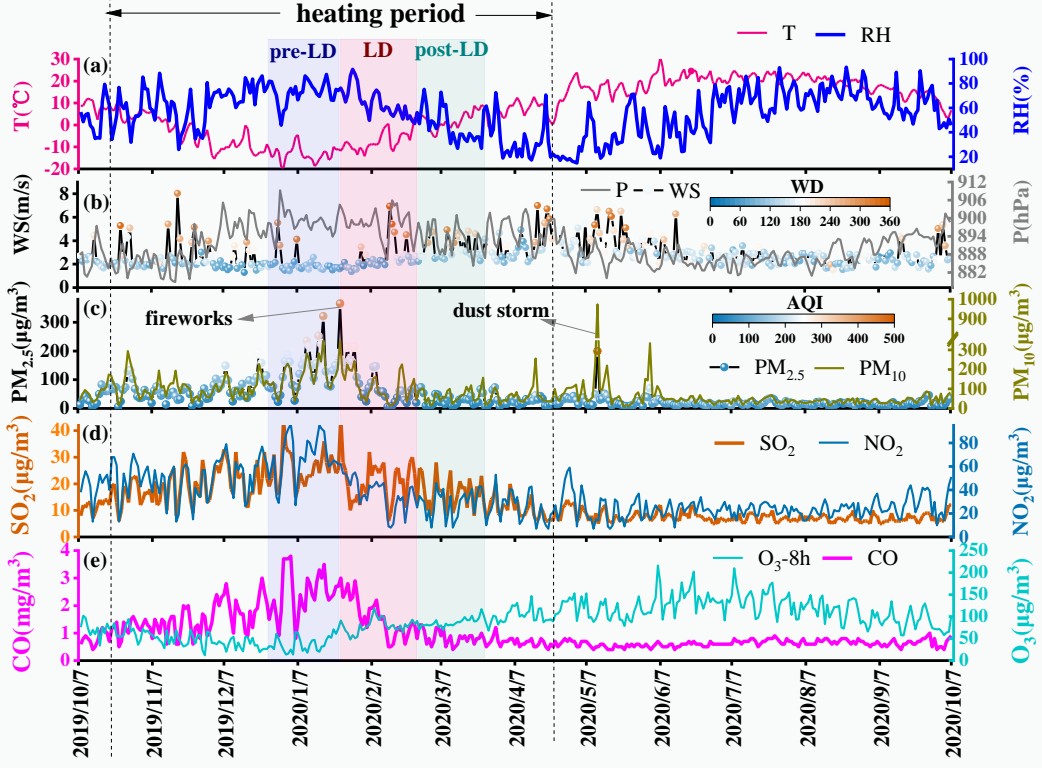

**Figure 2.** Daily variations in atmospheric pollutants and meteorological variables in Hohhot during the sampling period from 8$^{th}$ October, 2019 to 7$^{th}$ October, 2020. The blue, red, and green backgrounds represent the pre-lockdown, lockdown, and post-lockdown periods, respectively. T, RH, WS, WD, P, and AQI represent the ambient temperature, relative humidity, wind speed, wind direction, atmospheric pressure, and air quality index, respectively.

During haze episodes, a substantial increase in secondary inorganic components (sulfate, nitrate, and ammonium, SNA) was observed (Figure 3a–3c). The rapid increase in SNA was the main driving factor behind the increase in $PM_{2.5}$. High RH is conducive to the secondary formation of sulfates and nitrates, presenting higher SOR and NOR in these pollution periods (Figure 3a, 3b). In the heating period, in addition to the contribution of SNA to $PM_{2.5}$, the primary pollutants such as $Cl^-$ (*p < 0.001*) and EC (*p < 0.001*) were higher than those in the non-heating period (Figure 3d, 3e). Hohhot is an inland city, basically unaffected by sea salt. Furthermore, a higher average $Cl^-/Na^+$ ratio (3.43 for January) suggests the presence of non-marine anthropogenic sources of chloride. In conclusion, chloride is mainly emitted from coal combustion facilities in Hohhot, especially during the heating period. It can be used as an auxiliary marker of coal combustion in Hohhot. Higher SOR was observed in winter and summer in Hohhot (Figure 3a). High SOR in winter is mainly caused by heterogeneous processes under high RH conditions, while that in summer is caused by homogeneous gas-phase oxidation reactions under high temperatures and $O_3$ concentrations (Zhang et al., 2018; Li et al., 2017a). NOR was higher in winter, whereas it was lower in summer (Figure 3b). The higher NOR in winter can be ascribed to the rapid formation of nitrate under high RH (Fu et al., 2022; Ren et al., 2022). The lower NOR in summer may be related to the high temperature, which is favorable for nitrate volatilization (Daher et al., 2012). The higher SOR and NOR in winter indicate the higher secondary formation of sulfate and nitrate that resulted the heavy pollution episodes in pre-LD and LD periods.

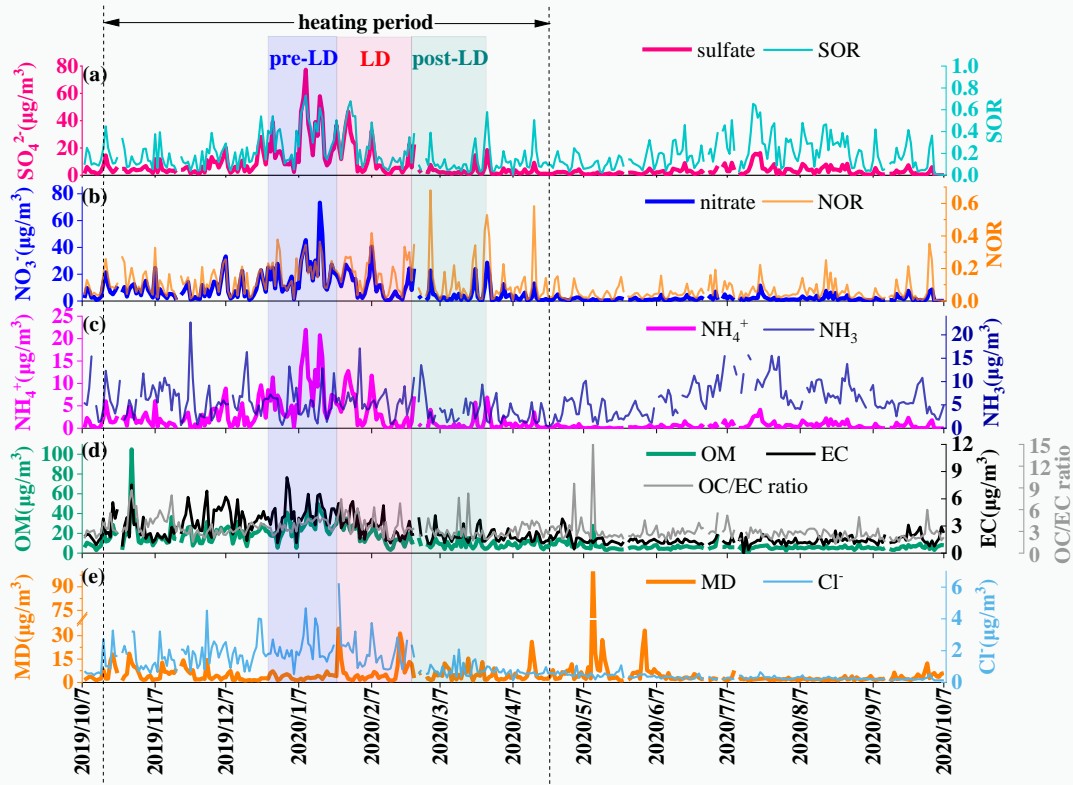

**Figure 3.** Daily variations in chemical composition of $PM_{2.5}$, SOR, NOR, $NH_3$, and OC/EC ratio in Hohhot during the sampling period. The blue, red, and green background represent the pre-lockdown, lockdown, and post-lockdown periods, respectively. OM, MD, SOR, and NOR represent the organic matter, mineral dust, sulfur oxidation ratio, and nitrogen oxidation ratio, respectively.

The comparison of atmospheric pollutants ($PM_{2.5}$, $PM_{10}$, $SO_2$, $NO_2$, $O_3$, and CO) between the LD period and the same period in 2017–2019 are shown in Figure S1. The average concentrations of $PM_{2.5}$, $PM_{10}$, $O_3$, and CO increased by 77.8% ($p < 0.01$), 34.6% ($p < 0.05$), 14.5% ($p < 0.001$), and 5.9% ($p < 0.001$), respectively, whereas the average concentrations of $SO_2$ and $NO_2$ decreased by 43.2% ($p < 0.05$) and 8.6% ($p < 0.001$), respectively.

The annual mean concentration of OM, $SO_4^{2-}$, $NO_3^-$, MD, EC, $NH_4^+$, and $Cl^-$ were 12.1, 6.6, 6.4, 4.9, 2.2, 2.0, and 1.1μg/m$^3$ (Figure 4a), accounting for 31.5%, 13.4%, 12.3%, 14.2%, 6.6%, 3.3%, and 2.5% of $PM_{2.5}$, respectively (Figure 4b). Compared with the results of Hohhot in 2014－2015 (Wang et al., 2019), the annual mean concentration of $NO_3^-$ increased, whereas the concentration of the other species decreased (Figure S10a). Due to the implemented measures, a sharp decrease in OM and MD was observed, resulting in a considerable decrease in $PM_{2.5}$ (decreased from 66μg/m$^3$ in 2014－2015 to 42.6μg/m$^3$ in 2019－2020). The proportion of $SO_4^{2-}$, $NO_3^-$, and OM increased considerably, whereas the proportion of MD showed a substantial decrease (Figure S10b). The result suggeststhat the contribution of chemical composition related to secondary formation has increased in recent years. However, the proportion of MD was still substantially higher than those of other cities in South China (Huang et al., 2013a), southwest China (Feng et al., 2021), southeast China (Li et al., 2017b), and the Central Plains Urban Agglomeration (Liu et al., 2019), which is close to the cities in northern China (Liu et al., 2021;

Xie et al., 2019) and northwest China (Zhou et al., 2021). The lowerRH, higher WS, and larger area of uncovered surface soil lead to frequent dust storms in semi-arid regions, resulting in a higher contribution of MD than in the humid area (Liang et al., 2019). The result indicates that the cities in arid or semi-arid regions (such as in northern China and northwest China) are more susceptible to mineral dust sources. The monthly average concentrations of SNA and OM during the heating period ($15^{th}$ October to $15^{th}$ April next year), especially in January, were higher than those of other non-heating months (Figure 4a), which were related to the coupled effect of a large amount of atmospheric precursors ($SO_2$, $NO_2$, and volatile organic compounds) and unfavorable meteorological conditions (high RH and low WS; Figure S11). Due to the frequent dust storms, the average concentration of MD in May (9.4 μg/m$^3$) was considerably higher than that of other months, accounting for 21.8% of $PM_{2.5}$ mass. The relatively high proportion of sulfates in August may be caused by its higher SOR, which is enhanced by photochemistry under high T, strong solar radiation, and high RH conditions.

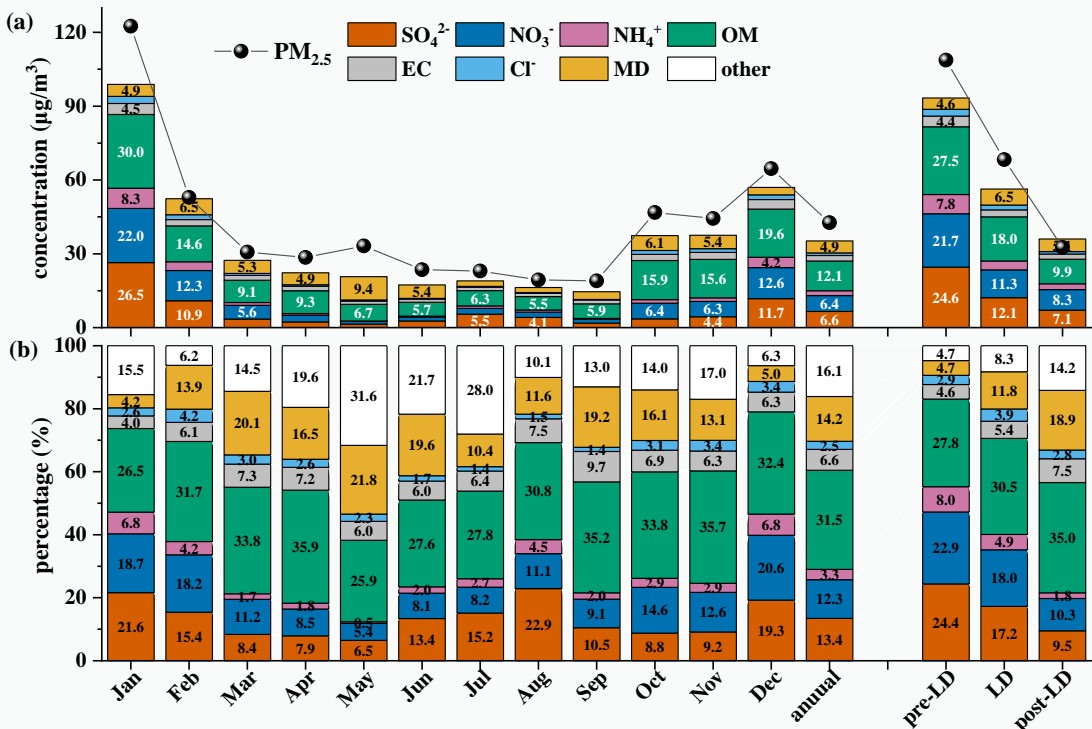

**Figure 4.** Monthly variation in (a) concentrations and (b) percentages of chemical components of $PM_{2.5}$ in Hohhot during the sampling period. Pre-LD, LD, and post-LD represent the pre-lockdown, lockdown, and post-lockdown periods, respectively. OM and MD represent organic matter and mineral dust, respectively.

The OM, sulfate, nitrate, and ammonium were the predominant components of $PM_{2.5}$ during the pre-LD period, accounting for 27.8, 24.4, 22.9, and 8.0% of $PM_{2.5}$ mass, respectively (Figure 4). During this period, the SNA contributed 55.3% to the total $PM_{2.5}$, slightly higher than those of the cities in northern China such as Xi'an (50.0%) (Tian et al., 2021) and Beijing (48.5%) (Ren et al., 2021), and lower than

the cities in southern China such as Guangzhou (78.7%) (Wang et al., 2021), Nanjing (68.2%) (Ren et al., 2021), and Shanghai (75.4%) (Chen et al., 2020). Sulfate was the predominant component of SNA in Hohhot during this period, whereas nitrite was the main contributor to SNA in Guangzhou, Nanjing, and Shanghai. The result indicated that higher SNA contributions in megacities of southern China are mainly related to vehicular emission. The higher contribution of sulfate in Hohhot is mainly related to coal combustion for winter heating. OM contributed 27.8% to the total $PM_{2.5}$, lower than that of Xi'an (42.0%) (Tian et al., 2021), and higher than that of the other cities listed in Table S9. The contribution of EC is higher than all of the cities listed in Table S9. The higher contribution of sulfate, OM and EC in Hohhot indicated that coal combustion may have been a predominant source of $PM_{2.5}$ during the pre-LD period. Most studies listed in Table S9 used online data, from which it is not possible to calculate the contribution of MD. However, our offline data showed that MD contributed 11.8% and 14.2% to the total $PM_{2.5}$ during the LD period and for the whole sampling year, respectively, indicating that MD is one of the main contributors of $PM_{2.5}$ that has been neglected in previous studies. The proportion of chemical species ranked as pre-LD: OM (27.8%) > $SO_4^{2-}$ (24.4%) > $NO_3^-$ (22.9%) > $NH_4^+$ (8.0%) > MD (4.7%) > EC (4.6%) > $Cl^-$ (2.9%); LD: OM (30.5%) > $NO_3^-$ (18.0%) > $SO_4^{2-}$ (17.2%) > MD (11.8%) > EC (5.4%) > $NH_4^+$ (4.9%) > $Cl^-$ (3.9%); and post-LD: OM (35.0%) > MD (18.9%) > $NO_3^-$ (10.3%) > $SO_4^{2-}$ (9.5%) > EC (7.5%) > $Cl^-$ (2.8%) > $NH_4^+$ (1.8%). Compared with the pre-LD period, the concentration of sulfate ($p < 0.01$), nitrate ($p < 0.01$), ammonium ($p < 0.01$), OM ($p < 0.001$), and EC ($p < 0.001$) decreased substantially due to the decline in the emission intensity under the strict control measures during the LD period (Figure 4a, Table S10). The percentage of sulfate (not significant for LD and $p < 0.01$ for post-LD), nitrate (not significant for LD and $p < 0.05$ for post-LD), and ammonium ($p < 0.05$ for LD and $p < 0.01$ for post-LD) decreased continuously during LD and post-LD, while the MD ($p < 0.01$ for LD and $p < 0.001$ for post-LD), OM (not significant for both two periods), and EC (not significant for LD and $p < 0.01$ for post-LD) increased (Table S10). The mean value of RH declined continuously from pre-LD to LD and post-LD, while the mean value of WS showed an opposite trend (Figure S11). The lower RH and higher WS were not conducive to the secondary formation and accumulation of SNA. Therefore, due to the emission reduction and improved atmospheric conditions, the proportion of SNA decreased sufficiently (from 55.3% in pre-LD to 40.1% in LD and 21.6% in post-LD). The atmospheric diffusion conditions improved during the post-LD period, and the concentration of OM, SNA, EC, and $Cl^-$ decreased substantially. These results suggest that the substantial changes that occurred in source contributors after the COVID-19 outbreak resulted in dramatic changes to aerosol composition.

To elucidate the rapid increase in $PM_{2.5}$, the sampling days were divided into four categories according to the daily concentration of $PM_{2.5}$: clean (CP, $PM_{2.5} < 35$ μg/m³), slightly polluted (SP, $35 \leq PM_{2.5} < 75$ μg/m³, moderately polluted (MP, $75 \leq PM_{2.5} < 150$ μg/m³, and heavily polluted (HP, 150μg/m³ $\leq PM_{2.5}$). The values of 24[th] January, 2020 (Chinese New Year's Eve) and 11[th] May, 2020 (a dust storm day) were excluded from the HP analysis. These two heavy pollution days were analyzed separately as two types, namely fireworks and dust storm. The meteorological conditions, gaseous precursors, and chemical composition of different pollution levels and types are shown in Figure 5. The concentrations of OM, sulfate, nitrate, and ammonium were in the order of CP < SP < MP < HP.

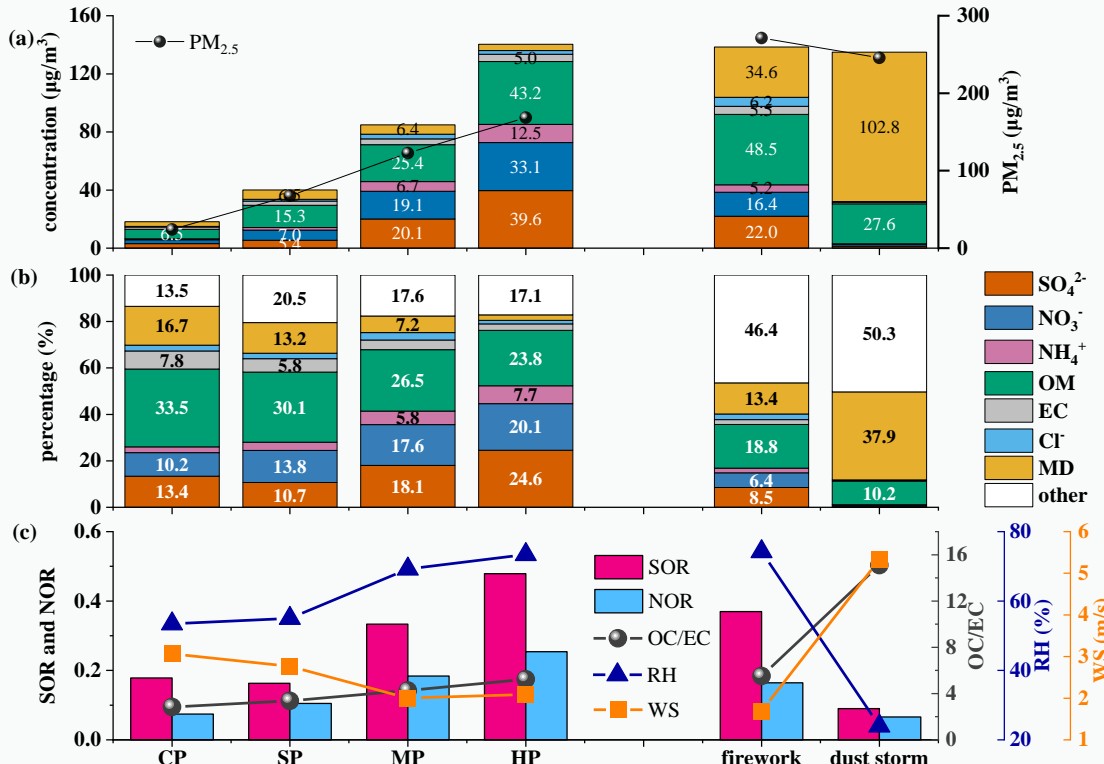

**Figure 5.** (a) Concentrations and (b) percentages of chemical components in $PM_{2.5}$, (c) meteorological conditions, SOR, NOR, and OC/EC at different pollution levels and during different types of pollution events. CP, SP, MP, and HP represent the clean ($PM_{2.5} < 35$ μg/m³), slightly polluted ($35 \leq PM_{2.5} < 75$ μg/m³), moderately polluted ($75 \leq PM_{2.5} < 150$ μg/m³), and heavily polluted periods ($PM_{2.5} \geq 150$ μg/m³ ), respectively.

From CP to HP, the percentages of SNA increased (from 26.1% to 52.4%), whereas the percentages of OM and MD decreased (from 33.5 to 23.8%, and from 16.7 to 2.4%, respectively). This response is related to the adverse meteorological conditions characterized by high RH and low WS, leading to the enhanced formation of SNA (higher SOR and NOR). The values of SOR increased from 0.18 during CP to 0.48 during HP. The values of NOR increased from 0.07 during CP to 0.25 during HP. The results suggest enhanced SNA formation during heavy pollution episodes. The coupled effects of high RH and low WS promoted the rapid increase of fine particulate matter on haze days in Hohhot. The high WS is beneficial for the elimination of atmospheric pollutants, resulting in low concentrations of $SO_2$ and $NO_2$ on dust storm days. Furthermore, the low RH is detrimental to the secondary formation of SNA (lower SOR and NOR), resulting in a lower SNA content in dust storm days. MD and OM contribute 102.8 and 27.6 μg/m³ to $PM_{2.5}$ during dust storm days, accounting for 37.9% and 10.2% of $PM_{2.5}$, respectively. The proportion of MD was highest in dust storm days, mainly because of the relatively high WS and low RH that were conducive to the re-suspension of crustal dust. During Lunar New Year, fireworks discharge a large number of gaseous pollutants, coupled with low WS and high RH, the concentrations of SNA, OM, and EC increased rapidly, resulting in serious pollution.

## 3.2 Factors influencing PM$_{2.5}$

The correlations between the chemical composition of PM$_{2.5}$, meteorological variables, and air pollutants are shown in Figure 6. PM$_{2.5}$ was negatively correlated with O$_3$, T, and WS at $p < 0.001$, indicating that high WS was beneficial for the elimination of fine particulate matter, while O$_3$ and T were mainly related to seasonal variation in sources and meteorological conditions. PM$_{2.5}$ was positively correlated with most of the aerosol components and gaseous pollutants, indicating that the source of PM$_{2.5}$ was very complex and influenced by a variety of factors. The SNA in PM$_{2.5}$ was positively correlated with RH ($p < 0.001$), indicating that high RH promotes heterogeneous formation of SNA.

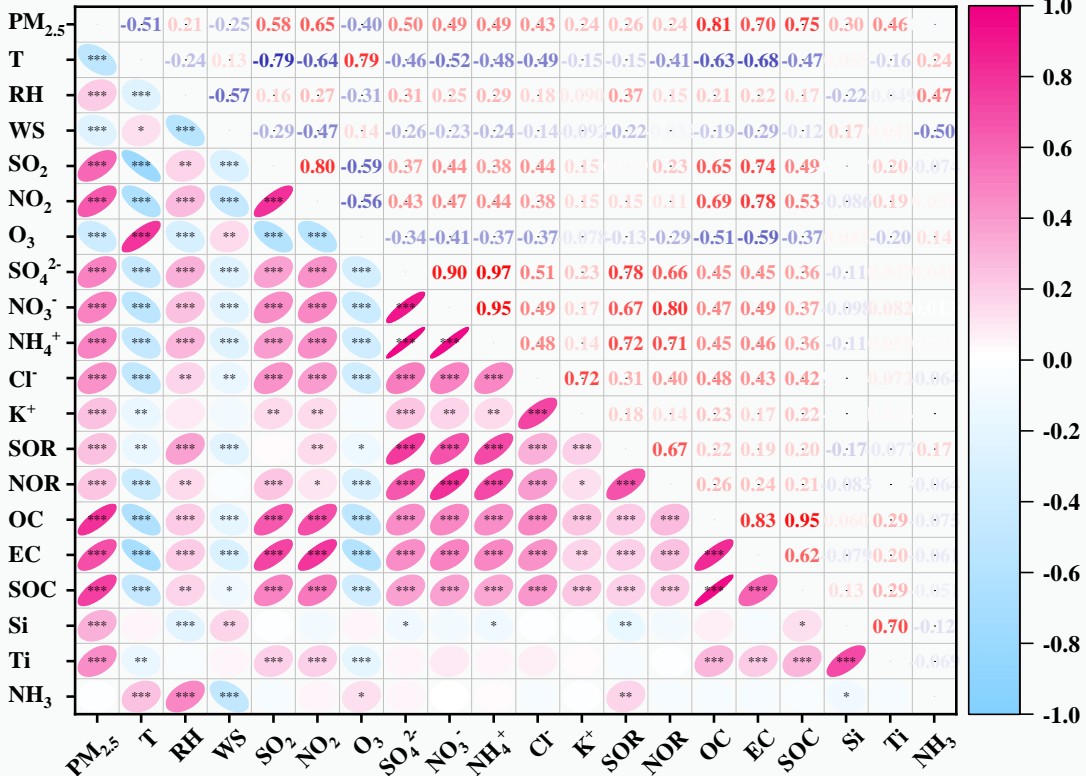

**Figure 6.** Correlation between chemical components of PM$_{2.5}$, meteorological variables, SOR, NOR, and air pollutants in Hohhot during the sampling period (* $p < 0.05$,** $p < 0.01$,*** $p < 0.001$).

The results suggest that RH played a vital role in the formation of haze by accelerating the conversion of SO$_2$ to SO$_4^{2-}$ and NO$_2$ to NO$_3^-$, deteriorating the air quality. NOR was negatively correlated with T at $p < 0.001$, which may be related to the volatility of NH$_4$NO$_3$. The higher T is favorable for nitrate volatilization, resulting in lower NOR (He et al., 2012). SOR and NOR were positively correlated with RH at $p < 0.001$ and $p < 0.05$, respectively, suggesting that both SOR and NOR were influenced by RH. The higher transformation of SO$_2$ to SNA was negatively correlated with WS ($p < 0.001$), indicating that high WS was conducive to the rapid elimination of SNA. SNA and its gaseous precursors (SO$_2$ and NO$_2$) were positively correlated ($p < 0.001$) but not related to NH$_3$, indicating that the formation of SNA was mainly controlled by SO$_2$ and NO$_2$ rather than NH$_3$. The carbonaceous aerosols (OC, EC, and SOC)

were positively correlated with Cl$^-$, SNA, $SO_2$, and $NO_2$ ($p < 0.001$), which are mainly affected by their common source in coal, used in heating. Silicon was positively correlated with WS ($p < 0.01$), indicating that high WS is beneficial to the re-suspension of soil or dust, resulting in an increase in Si in $PM_{2.5}$. Silicon was negatively correlated with RH ($p < 0.001$), which was related to high WS and low RH in dust storm days.

### 3.3 Source apportionment of $PM_{2.5}$

The sources of $PM_{2.5}$ were apportioned using the PMF model (EPA PMF 5.0). The results of the source apportionment during different sampling periods are shown in Figure 7 and summarized in Table S11. The $PM_{2.5}$ concentrations in spring, summer, autumn, winter, and annually were 32.4, 24.3, 37.0, 80.8, and 42.6 $\mu g/m^3$, respectively. Coal combustion (CC), vehicular exhaust (VE), crustal sources (CS), and secondary inorganic aerosols (SIA) were the main contributors to $PM_{2.5}$ over the sampling year, contributing 38.3%, 35.0%, 13.5%, and 11.4% to $PM_{2.5}$, respectively. The contribution of primary sources such as CC, VE, and dust source (refer to the sum of construction dust and crustal sources in this study) in Hohhot was higher than the megacities such as Beijing (Zíková et al., 2016), Tianjin (Tian et al., 2021), and Shanghai (Feng et al., 2022a), whereas the SIA and biomass burning (BB) contributions were lower than in these cities (Table S11). The result indicates that the contribution of secondary aerosols is predominant in megacities, while the primary source is predominant in semi-arid regions. Therefore, the control of primary sources is an effective way to reduce the concentration of $PM_{2.5}$ in Hohhot.

The CC contribution to $PM_{2.5}$ in spring, summer, autumn, and winter was 14.6, 5.7, 12.4, and 41.3 $\mu g/m^3$, with a contribution percentage of 56.1%, 24.0%, 38.9%, and 65.4%, respectively. Coal combustion was the main contributor to $PM_{2.5}$ in Hohhot, especially during the heating period. Summer is the only season for completely no coal-fired heating, a relatively low contribution of CC was observed in summer. The VE contribution concentrations in spring, summer, autumn, and winter were 4.4, 11.5, 10.7, and 9.0 $\mu g/m^3$, contributing 17.0%, 48.4%, 33.8%, and 14.3% to $PM_{2.5}$, respectively. The peak seasonal contribution percentage of VE was observed in summer. This is mainly attributable to a substantial decline in the contribution of other sources, increasing the proportion of VE. The contribution concentration of SIA followed the order of winter (6.6 $\mu g/m^3$) > autumn (3.5 $\mu g/m^3$) > summer (1.2 $\mu g/m^3$) > spring (1.1 $\mu g/m^3$), with a contribution percentage of 10.5%, 11.1%, 5.3%, and 4.2%, respectively. The higher contribution of SIA can be attributed to the large amount of gaseous precursors emitted by CC in winter, whereas the higher SIA contribution in autumn was related to the high oxidation rate. A relatively low contribution was observed in spring. The lower contribution of SIA in spring may be related to the high WS and low RH, which is unfavorable for SNA formation and accumulation.

The contributions concentrations of CS followed the order of spring (4.9 $\mu g/m^3$) > autumn (4.4 $\mu g/m^3$) > winter (4.3 $\mu g/m^3$) > summer (3.2 $\mu g/m^3$), with a contribution percentage of 18.6%, 13.8%, 6.8%, and 13.5%, respectively. A relatively high contribution of CS to $PM_{2.5}$ was observed in spring, which is associated with the increased long-range transportation of crustal sources due to dry and windy weather in Hohhot. The higher contribution of dust sources to $PM_{2.5}$ has been reported in some other

semi-arid regions, such as Guanzhong basin (Li et al., 2022) and Lanzhou (Liang et al., 2019), indicating that the semi-arid regions are more susceptible to dust sources. The source apportionment results indicate that primary sources such as CC, VE, and dust sources in Hohhot were predominant, which is different from cities with high secondary pollution.

During the LD period, the contribution of SIA, CC, CS, BB, CD, and VE was 22.6, 18.2, 7.7, 5.6, 3.0, and 2.6 $\mu g/m^3$ to $PM_{2.5}$, respectively, accounting for 37.8%, 30.5%, 12.9%, 9.4%, 5.1%, and 4.4% of the total $PM_{2.5}$ mass (Figure 7). The contribution of CC and dust source (the sum of CS and CD) during the LD period in Hohhot was much higher than those of Tangshan (Wang et al., 2021), Taiyuan (Wang et al., 2022), and Xiamen (Hong et al., 2021) (Table S11). The contribution of SIA was lower than Tangshan and Taiyuan, while higher than Xiamen. Hohhot, Tangshan, and Taiyuan are located in northern China, and consume a large amount of coal for winter heating. The high intensity of gaseous precursors emitted from coal combustion is reasonable for a high contribution of SIA. The contribution of VE in Hohhot was lower than Xiamen and Taiyuan. The contribution of VE decreased from 35.5% to 4.4%, whereas the SIA increased from 21.1 % to 37.8 %. The substantial reduction in VE was associated with the strict traffic restrictions during the LD period, which is consistent with the findings in Taiyuan (Wang et al., 2022). Compared with the LD period, the contribution of VE increased from 4.4% to 14.7% during the post-LD period, which can be ascribed to the canceled traffic restrictions. The contribution of CC increased from 30.5% during the LD period to 68.7% during the post-LD period, while the concentration decreased from 29.2 to 18.2 $\mu g/m^3$. The contribution of SIA decreased from 37.8% during the LD period to 5.0% during the post-LD period, which can be attributed to the improved atmospheric conditions.

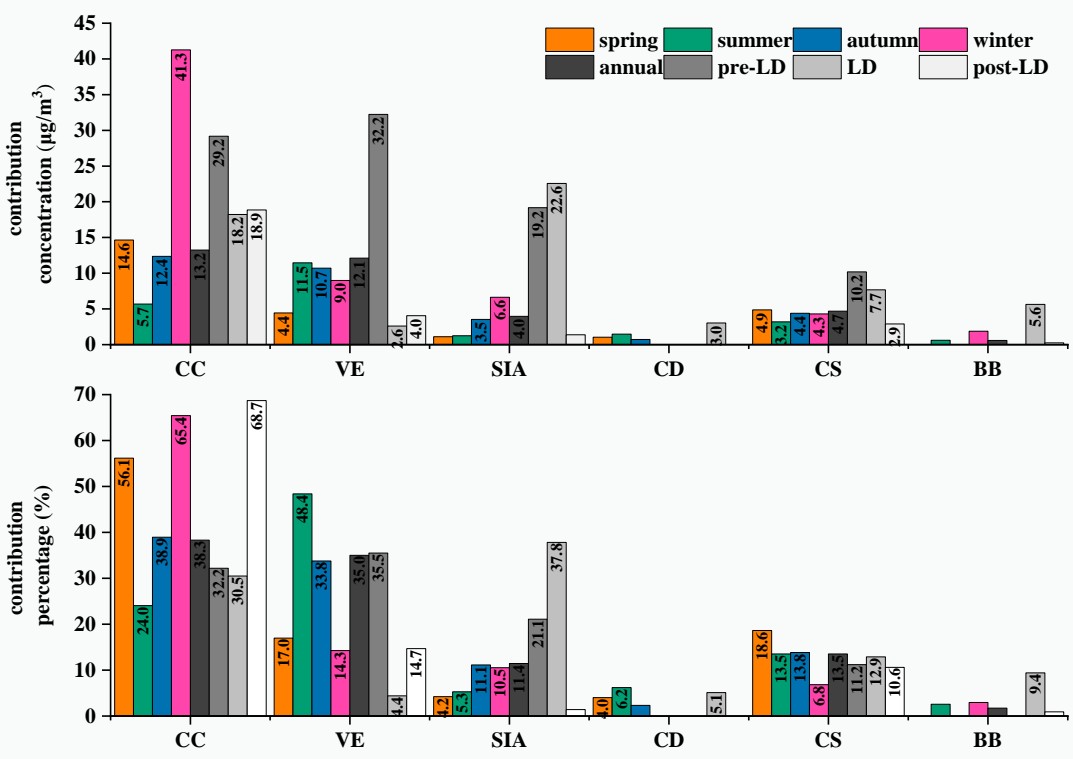

**Figure 7.** (a) Concentration and (b) percentage of source contribution to $PM_{2.5}$ in Hohhot in spring, summer, autumn, winter, over the sampling year, pre-lockdown, lockdown, and post-lockdown periods. CC, VE, SIA, CD, CS, and BB represent coal combustion, vehicular emission, secondary inorganic aerosol, construction dust, crustal sources, and biomass burning, respectively.

**4 Conclusion**

A single year's offline measurement was conducted in Hohhot to reveal the chemical characteristics and sources of $PM_{2.5}$ in a semi-arid region. Organic matter, mineral dust, sulfate, and nitrite were the predominant components of $PM_{2.5}$ in Hohhot, and coal combustion, vehicular emission, crustal sources, and secondary inorganic aerosols were the main contributors to $PM_{2.5}$. The high proportion of mineral dust composition and higher contribution of crustal sources to $PM_{2.5}$ indicated that cities in semi-arid regions are more susceptible to dust sources. The heavy pollution in winter can be attributed to the rapid increase of SNA under high RH and low WS conditions, while the heavy pollution in spring was associated with long-range transmission of crustal sources due to the dry and windy weather. Compared with the pre-LD period, the concentration of SNA, OM, and EC decreased substantially during LD and post-LD periods due to the lockdown measures. The source contribution of secondary inorganic aerosols and vehicular emission decreased during the lockdown period, whereas coal combustion increased. The substantial reduction in the contribution of vehicular emissions was associated with the strict traffic restrictions during the lockdown period, the increase in vehicular emission contributions during the post-lockdown period can be attributed to the canceled traffic restrictions.

A relatively high contribution of primary sources, such as coal combustion and dust source, was observed in Hohhot. Therefore, the control of primary sources, such as increasing the proportion of clean energy to reduce coal consumption, could be an effective way to reduce the concentration of $PM_{2.5}$ in Hohhot. The unfavorable meteorological conditions played an integral role during winter and promoted SNA formation and accumulation, causing frequent heavy pollution events. The reduction in anthropogenic activities and the important role of meteorology in the formation of air pollutants should be considered in aerosol quality and policy measures. The emission reduction of gaseous precursors ($SO_2$ and $NO_x$) under adverse meteorological conditions can prevent heavy pollution events driven by SNA. The control of coal combustion sources and accurate ambient air quality forecasting techniques will do much good to reduce annual concentrations of $PM_{2.5}$ and the occurrence of heavy pollution days, respectively. This study provides new insight for the formulation of effective policies to improve aerosol pollution in semi-arid regions.

*Data availability.* Data are available from the corresponding author upon request (hjzhou@imnu.edu.cn).

*Supplement.* The Supplement related to this article is available online at

*Author Contributions.* HJZ designed the study and prepared the paper with inputs from all the coauthors. Data

analysis and source apportionment were done by HJZ and TL. PL, JWW, and DDGL carried out the experiments.
YLT provided the air quality data. FH, BS, and XJZ participated in the field campaign and data analysis. XC and ZQW supervised the study.

*Competing interest.* The authors declare that they have no known competing financial interests or personal relationships that could have appeared to influence the work reported in this paper.

**Acknowledgments.** This work was supported by the National Natural Science Foundation of China
(42167015), Natural Science Foundation of Inner Mongolia (2018MS02001), Scientific Research Foundation for the High-level Talents of Inner Mongolia (24), and Scientific Research Foundation for the High-level Talents of Inner Mongolia Normal University (2018YJRC005), Science and Technology Project of Inner Mongolia Autonomous Region (2021GG0406).

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
