# Peer review of "Chemical Characteristics and Source of PM2.5 in Hohhot, a Semi-arid City in Northern China: Insight from the COVID-19 Lockdown"

_Atmospheric Chemistry and Physics, 2022_

## Author Comment (AC1)

**Responses to reviewer #1**

We appreciate the reviewer for the constructive comments on our manuscript. We have studied the comments carefully and revised our manuscript accordingly, which can be found in the attached file (Track Changes). Our point-by-point replies to the comments are provided below. Referee comments are given in black, and our replies are given in blue. Additionally, we checked our figures using the Coblis – Color Blindness Simulator and revised the color schemes accordingly.

**Comments**

The manuscript addresses a topic of scientific interest during the last 2 years, such as the variation in air pollution during the COVID-19 lockdown. The study is interested in analyzing the variation of the chemical composition of $PM_{2.5}$ (not only its concentration) obtained in an area with particular geographical conditions such as a semi-arid city of northern China. Several studies have reported changes in the concentrations of atmospheric pollutants such as PM, $O_3$ and $NO_2$ during the lockdown measures, but few studies have delved into the variation in the chemical composition of PM. This approach allows carrying out more detailed analyzes of atmospheric chemistry by relating the fluctuation of emission sources and the implications on the chemical composition of PM.

I appreciate if the authors can offer a response/discussion to each of the following comments:

1. The authors define as objective of the study "identify the long-term chemical characteristics of $PM_{2.5}$ in a semi-arid city". However, can one year of study be considered a long-term study?

   **Response**: Thank you for pointing it out. We revised the inappropriate sentence. (L92)

2. The manuscript suggests that the results obtained "can provide a new insight for the formulation of effective policies to improve aerosol pollution in semi-arid regions". The authors should go beyond the generality and could suggest concrete measures to improve public policies based on the results achieved.

   **Response**: Thank you for your suggestion. We add some practical suggest on improving air quality in the study area, which also can be applied in semi-arid region.

   L422-433:*"A relatively high contribution of primary sources, such as coal combustion and dust source, was observed in Hohhot. Therefore, the control of primary sources, such as*

*increasing the proportion of clean energy to reduce coal consumption, could be an effective way to reduce the concentration of $PM_{2.5}$ in Hohhot. The unfavorable meteorological conditions played an integral role during winter and promoted SNA formation and accumulation, causing frequent heavy pollution events. The reduction in anthropogenic activities and the important role of meteorology in the formation of air pollutants should be considered in aerosol quality and policy measures. The emission reduction of gaseous precursors ($SO_2$ and $NO_x$) under adverse meteorological conditions, can prevent heavy pollution events driven by SNA. The control of coal combustion sources and accurate ambient air quality forecasting techniques will do much good to reduce annual concentrations of $PM_{2.5}$ and the occurrence of heavy pollution days, respectively. This study can provide a new insight for the formulation of effective policies to improve aerosol pollution in semi-arid regions. "*

3. It would be interesting to present a comparative analysis of the variation in the composition of $PM_{2.5}$ (not only concentrations) between the year of study and an average of previous years (to be possible). This is a good way to identify $PM_{2.5}$ chemical composition anomalies during the COVID-19 lockdown measures.

**Response**: We add a comparison of chemical composition of $PM_{2.5}$ between previous study (in 2014-2015) (Wang et al., 2019) and our result (in 2019-2020).

L223-239: *"The annual mean concentration of OM, $SO_4^{2-}$, $NO_3^-$, MD, EC, $NH_4^+$, and $Cl^-$ were 12.1, 6.6, 6.4, 4.9, 2.2, 2.0, and 1.1μg/m$^3$ (Figure 4a), accounting for 31.5%, 13.4%, 12.3%, 14.2%, 6.6%, 3.3%, and 2.5% of $PM_{2.5}$, respectively (Figure 4b). Compared with the result of Hohhot in 2014–2015(Wang et al., 2019), the annual mean concentration of $NO_3^-$ increased, whereas the concentration of the other species decreased (Figure S10a). Due to the implemented measures, a sharp decrease in OM and MD was observed, resulting in a considerable decrease in $PM_{2.5}$ (decreased from 66μg/m$^3$ in 2014–2015 to 42.6μg/m$^3$ in 2019–2020). The proportion of $SO_4^{2-}$, $NO_3^-$, and OM increased considerably, whereas the proportion of MD showed a substantial decrease (Figure S10b). The result indicates that the contribution of chemical composition related to secondary formation has increased in recent years. However, the proportion of MD was still substantially higher than those of other cities in South China (Huang et al., 2013), southwest China (Feng et al., 2021), southeast China (Li et al., 2017b), and the Central Plains Urban Agglomeration (Liu et al., 2019), which is close to the cities in northern China (Liu et al., 2021; Xie et al., 2019) and northwest China (Zhou et al., 2021). The lower relative humidity, higher wind speed, and larger area of uncovered surface soil lead to frequent dust storms in semi-arid regions, resulting in a higher contribution of MD than in the humid area. The result indicates that the cities in arid or semi-arid regions (such as in northern China and northwest China) are more susceptible to mineral dust sources."*

[Figure]

*Figure S3.* Comparison of chemical composition in (a) 2019-2020 and (b) 2014-2015 in Hohhot. The data of chemical composition in 2014-2015 were obtained from Wang (Wang et al., 2019). The organic matter (OM) and mineral dust (MD) were calculated by the following equations with the composition data, $OM=1.6\times[OC]$ and $MD=2.14\times[Si]+1.89\times[Al]+1.40\times[Ca]+1.43\times[Fe]+1.58[Mn]+1.21\times[K]+1.67\times[Ti]$.

4. The authors calculated and reported two indicators related to secondary aerosols, the sulfur oxidation ratio (SOR) and the nitrogen oxidation ratio (NOR). What is the usefulness of these indicators and how are the results interpreted? What additional information do the indicators provide regarding the concentrations of $SO_2$ and $SO_4$?

**Response**: The sulfur oxidation ratio (SOR) and nitrogen oxidation ratio (NOR) are important indicators to estimate the secondary formation of inorganic aerosols. The increasing SOR and NOR were observed during heavy pollution process, indicating that the secondary formation of sulfate and nitrate is the main driving factor for the rapid increase of fine particulate matter on haze days in Hohhot. We revised section 2.3 and the related text in our manuscript.

L142-145: *"The organic matter (OM) and mineral dust (MD) were calculated using the following equations (1 - 2). To estimate the secondary formation of inorganic and organic aerosols, the sulfur oxidation ratio (SOR), nitrogen oxidation ratio (NOR), and secondary organic carbon (SOC) were calculated using the following equations (3 - 5) (Xie et al., 2019; Liu et al., 2021)"*

L186-187: *"High RH is conducive to the secondary formation of sulfates and nitrates, presenting higher SOR and NOR in these pollution periods (Figure 3a, 3b)."*

L192-200: *"Higher SOR was observed in winter and summer in Hohhot (Figure 3a). High SOR in winter is mainly caused by heterogeneous processes under high RH conditions, while that in summer is caused by homogeneous gas-phase oxidation reactions under high temperatures and $O_3$ concentrations (Zhang et al., 2018; Li et al., 2017a). NOR was higher in winter, whereas it was lower in summer (Figure 3b). The higher NOR in winter can be ascribed to the rapid formation of nitrate under high RH. The lower NOR in summer may be related to the high temperature, which is favorable for nitrate volatilization (Daher et al., 2012). The higher SOR and NOR in winter indicate the higher secondary formation of sulfate and nitrate that resulted the heavy pollution episodes in pre-LD and LD periods."*

L305-314: *"From CP to HP, the percentages of SNA increased (from 26.1% to 52.4%), whereas the percentages of OM and MD decreased (from 33.5% to 23.8%, and from 16.7 to 2.4, respectively). This response is related to the adverse meteorological conditions characterized by high RH and low WS, leading to the enhanced formation of SNA (higher SOR and NOR). The values of SOR increased from 0.18 during CP to 0.48 during HP. The values of NOR increased from 0.07 during CP to 0.25 during HP. The results suggest enhanced SNA formation during heavy pollution episodes. The coupled effects of high RH and low WS promoted the rapid increase of fine particulate matter on haze days in Hohhot. The high WS is beneficial for the elimination of atmospheric pollutants, resulting in low concentrations of $SO_2$ and $NO_2$ on dust storm days. Furthermore, the low RH is detrimental to the secondary formation of SNA (lower SOR and NOR), resulting in a lower SNA content in dust storm days."*

L334-338: *"NOR was negatively correlated with T at $p < 0.001$, which may be related to the volatility of $NH_4NO_3$. The higher T is favorable for nitrate volatilization, resulting in lower NOR (He et al., 2012). SOR and NOR was positively correlated with RH at $p < 0.001$ and $p < 0.05$, respectively, suggesting that both SOR and NOR were influenced by RH."*

5. Please check in the title "3.2 Factors influencing $PM_{2.5}$" the word "metrological" since it should be "meteorological".

**Response**: We checked through our manuscript and corrected the typos. (L321 , L330, and L425)

6. The source apportionment of PM$_{2.5}$ was carried out for each of the four seasons. But how did the COVID-19 lockdown measures impact on sources of PM$_{2.5}$?

**Response**: Thanks for your constructive comment. We conducted a study on the impact of COVID-19 lockdown measures on PM$_{2.5}$ sources. We added some discussions to the section 3.3 and made a revision on the conclusion section.

L383-399: *"During the LD period, the contribution of SIA, CC, CS, BB, CD, and VE was 22.6, 18.2, 7.7, 5.6, 3.0, and 2.6 μg/m³ to PM$_{2.5}$, respectively, accounting for 37.8%, 30.5%, 12.9%, 9.4%, 5.1%, and 4.4% of the total PM$_{2.5}$ mass (Figure 7). The contribution of CC and dust source (the sum of CS and CD) during the LD period in Hohhot was much higher than those of Tangshan (Wang et al., 2021), Taiyuan (Wang et al., 2022), and Xiamen (Hong et al., 2021) (Table S11). The contribution of SIA was lower than Tangshan and Taiyuan, while higher than Xiamen. Hohhot, Tangshan, and Taiyuan are located in northern China, and consume large amount of coal for winter heating. The high intensity of gaseous precursors emitted from coal combustion is reasonable for a high contribution of SIA. The contribution of VE in Hohhot was lower than Xiamen and Taiyuan. The contribution of VE decreased from 35.5% to 4.4%, whereas the SIA increased from 21.1 % to 37.8 %. The substantial reduction in VE was associated with the strict traffic restrictions during the LD period, which is consistent with the findings in Taiyuan (Wang et al., 2022). Compared with the LD period, the contribution of VE increased from 4.4% to 14.7% during the post-LD period, which can be ascribed to the canceled traffic restrictions. The contribution of CC increased from 30.5% during the LD period to 68.7% during the post-LD period, while the concentration decreased from 29.2 to 18.2 μg/m³. The contribution of SIA decreased from 37.8% during the LD period to 5.0% during the post-LD period, which can be attributed to the improved atmospheric conditions."*

L29-30: *"The contribution of secondary inorganic aerosols increased (from 21.1 to 37.8%), whereas the contribution of vehicular emissions was reduced (35.5% to 4.4%) due to lockdown measures."*

L417-421: *"The source contribution of secondary inorganic aerosols and vehicular emission decreased during the lockdown period, whereas coal combustion increased. The substantial reduction in the contribution of vehicular emissions was associated with the strict traffic restrictions during the lockdown period, the increase in vehicular emission contributions during the post-lockdown period can be attributed to the canceled traffic restrictions."*

[revised manuscript text omitted]

---

## Author Comment (AC2)

**Responses to reviewer #2**

We appreciate the reviewer for the constructive comments on our manuscript. We have studied the comments carefully and revised our manuscript accordingly, which can be found in the attached file (Track Changes). Our point-by-point replies to the comments are provided below. Referee comments are given in black, and our replies are given in blue. Additionally, we checked our figures using the Coblis – Color Blindness Simulator and revised the color schemes accordingly.

**Comments**

This manuscript analyses the composition and sources of ambient $PM_{2.5}$ in the Hohhot region in China before and during the COVID-19 lockdown. The information presented in the study is relevant because, unlike other existing studies, data were collected well before beginning and after the lockdown period which allowed capturing business-as-usual and lockdown $PM_{2.5}$ samples. Results are well presented and structured, and the discussion goes straight to the relevant findings. By applying the PMF model, it can be proposed those sources that contribute significantly to the ambient levels observed with statistical confidence. Nevertheless, some details must be addressed before it is accepted for publication at ACP.

1. Some sentences are repetitive in the abstract, e.g. L25-27, and within the entire document.

   **Response**: Thank you for pointing it out. We conducted a statistical test and revised the related descriptions.

   L26-29: We deleted the description of not-significant variation.

   L279-283: We rephrased the sentence.

   *"Compared with the pre-LD period, the concentration of sulfate (p<0.01), nitrate (p<0.01), ammonium (p<0.01), OM (p<0.001), and EC (p<0.001) decreased due to the decline in the emission intensity under the strict control measures during the LD period (Figure 4a, Table S10). The percentage of sulfate (not significant for LD and p<0.01 for post-LD), nitrate (not significant for LD and p<0.05 for post-LD), and ammonium (p<0.05 for LD and p<0.01 for post-LD) decreased continuously during LD and post-LD, while the MD (p<0.01 for LD and p<0.001 for post-LD), OM (not significant for both two periods), and EC (not significant for LD and p<0.01 for post-LD) increased (Table S10)."*

   L413-415: We rephrased the sentence.

   *"Compared with the pre-LD period, the concentration of SNA, OM, and EC decreased substantially during LD and post-LD periods due to the lockdown measures."*

2. One aim is to identify the long-term characteristics of PM$_{2.5}$ in the studied region, however, analysing one year is not sufficient to understand long-term variations unless their results are discussed and compared with those in existing studies, which are not reported.

   **Response**: Thank you for your suggestion. We revised the inappropriate sentence. (L92)

3. Introduction includes studies from most regions of the world, but Latin America was not included where also interesting studies have been made. I recommend you revise and include the following studies where appropriate:

   -Mendez-Espinosa, J. F., Rojas, N. Y., Vargas, J., Pachón, J. E., Belalcazar, L. C., & Ramírez, O. (2020). Air quality variations in Northern South America during the COVID-19 lockdown. Science of the Total Environment, 749, 141621.
   -Hernández-Paniagua, I. Y., Valdez, S. I., Almanza, V., Rivera-Cárdenas, C., Grutter, M., Stremme, W., García-Reynoso, A. & Ruiz-Suárez, L. G. (2021). Impact of the COVID-19 lockdown on air quality and resulting public health benefits in the Mexico City Metropolitan Area. Frontiers in public health, 9, 642630.
   -Nakada, L. Y. K., & Urban, R. C. (2020). COVID-19 pandemic: Impacts on the air quality during the partial lockdown in São Paulo state, Brazil. Science of the Total Environment, 730, 139087.

   **Response**: Thanks for your comment. The references have been added to the manuscript. (L72-73)

4. In line 129: the authors did not define what a strict analytical procedure is.

   **Response**: The detailed analytical procedures were reported in previous studies, and we conducted our analysis according to the referred methods strictly. We added some quality assurance descriptions to make our data reliable.

   L136-139: *"Field blank and replicate analyses were carried out once per 10 samples. The concentrations of field blanks were all lower than the method detection limits, and the relative deviations of replicate analyses were < ~ 5%. All the analytical procedures were strictly controlled according to the referred methods to reduce artificial interference."*

5. In several sections, calculations and results are reported for a year time-scale, but it is not defined if this refer to a calendar or sampling year.

   **Response**: All the years mentioned in the manuscript refer to the sampling year. We revised the ambiguous sentences. (L268, L352, and L402)

6. It would be convenient if the authors propose a hypothesis in introduction and then discuss their findings in light of it, e.g. L196-199.

**Response**: We revised it accordingly.

L77-80: *"An increase (p<0.01) in $PM_{2.5}$ was found in Hohhot during the LD period, whereas a considerable improvement was reported in most of the cities globally. The response of chemical composition and sources of $PM_{2.5}$ in Hohhot to lockdown measures and the driving factors behind the abnormal increase in $PM_{2.5}$ are still unclear."*

7. In most sections, $PM_{2.5}$ composition is claimed to be different from other Chinese regions but the reason behind this is not discussed. This issue is critical and must be addressed.

**Response**: Thanks for the comment. We added some discussions about the composition and source differences between Hohhot and other cities.

L232-237: **"***However, the proportion of MD was still substantially higher than those of other cities in South China (Huang et al., 2013), southwest China (Feng et al., 2021), southeast China (Li et al., 2017b), and the Central Plains Urban Agglomeration (Liu et al., 2019), which is close to the cities in northern China (Liu et al., 2021; Xie et al., 2019) and northwest China (Zhou et al., 2021). The lower relative humidity, higher wind speed, and larger area of uncovered surface soil lead to frequent dust storms in semi-arid regions, resulting in a higher contribution of MD than in the humid area. The result indicates that the cities in arid or semi-arid regions (such as in northern China and northwest China) are more susceptible to mineral dust sources.***"***

L254-265: **"***During this period, the SNA contributed 55.3% by to the total $PM_{2.5}$, slightly higher than those of the cities in northern China such as Xi'an (50.0%) (Tian et al., 2021) and Beijing (48.5%) (Ren et al., 2021), and lower than the cities in southern China such as Guangzhou (78.7%) (Wang et al., 2021), Nanjing (68.2%) (Ren et al., 2021), and Shanghai (75.4%) (Chen et al., 2020). Sulfate was the predominant component of SNA in Hohhot during this period, whereas nitrite was the main contributor to SNA in Guangzhou, Nanjing, and Shanghai. The result indicated that higher SNA contributions in megacities of southern China are mainly related to vehicular emission. The higher contribution of sulfate in Hohhot is mainly related to coal combustion for winter heating. OM contributed by 27.8% to the total $PM_{2.5}$, lower than that of Xi'an (42.0%) (Tian et al., 2021), and higher than that of the other cities listed in Table S9. The contribution of EC is higher than all of the cities listed in Table S9. The higher contribution of sulfate, OM and EC in Hohhot indicated that coal combustion may have been a predominant source of $PM_{2.5}$ during the pre-LD period.***"***

L353-358: *"The contribution of primary sources such as CC, VE, and dust source (refer to the sum of construction dust and crustal sources in this study) in Hohhot was higher than the megacities such as Beijing (Žková et al., 2016), Tianjin (Tian et al., 2021), and Shanghai (Feng et al., 2022), whereas the SIA and BB contributions were lower than in these cities*

*(Table S11). The result indicates that the contribution of secondary aerosols is predominant in megacities, while the primary source is predominant in semi-arid regions."*

8. L230-235: Statistical tests must be conducted to identify if the changes observed for each component between periods were significant.

**Response**: We conducted a statistical test for the changes of each component between different periods and marked the significance level to the text and figures. We added a table (Table S10) to the supplement.

L186-188: *"In the heating period, in addition to the contribution of SNA to $PM_{2.5}$, the primary pollutants such as $Cl^{-}$ (p<0.001) and EC (p<0.001) were higher than those in the non-heating period (Figure 3d, 3e)"*

L213-216: *"The comparison of atmospheric pollutants ($PM_{2.5}$, $PM_{10}$, $SO_2$, $NO_2$, $O_3$, and CO) between the LD period and the same period in 2017–2019 are shown in Figure S1. The average concentrations of $PM_{2.5}$, $PM_{10}$, $O_3$, and CO increased by 77.8% (p<0.01), 34.6% (p<0.05), 14.5% (p<0.001), and 5.9% (p<0.001), respectively, whereas the average concentrations of $SO_2$ and $NO_2$ decreased by 43.2% (p<0.05) and 8.6% (p<0.001), respectively."*

L274-283: *"Compared with the pre-LD period, the concentration of sulfate (p < 0.01), nitrate (p < 0.01), ammonium (p < 0.01), OM (p < 0.001), and EC (p < 0.001) decreased substantially due to the decline in the emission intensity under the strict control measures during the LD period (Figure 4a, Table S10). The percentage of sulfate (not significant for LD and p < 0.01 for post-LD), nitrate (not significant for LD and p < 0.05 for post-LD), and ammonium (p < 0.05 for LD and p < 0.01 for post-LD) decreased continuously during LD and post-LD, while the MD (p < 0.01 for LD and p < 0.001 for post-LD), OM (not significant for both two periods), and EC (not significant for LD and p < 0.01 for post-LD) increased (Table S10)."*

**Table S10** The changes of chemical composition of $PM_{2.5}$ in Hohhot during pre-LD, LD, post-LD

| species | period | Concentration | | Percentage | |
| --- | --- | --- | --- | --- | --- |
| | | change ($\mu g/m^3$) | p | Change (%) | p |
| $SO_4^{2-}$ | LD | -12.42 | 0.004 | -7.02 | 0.055 |
| | post-LD | -17.46 | 0.000 | -11.36 | 0.003 |
| $NO_3^-$ | LD | -10.38 | 0.007 | -4.75 | 0.210 |
| | post-LD | -13.41 | 0.001 | -8.23 | 0.036 |
| $NH_4^+$ | LD | -4.27 | 0.005 | -3.04 | 0.032 |
| | post-LD | -5.41 | 0.001 | -4.44 | 0.003 |
| Cl- | LD | -0.81 | 0.129 | +0.96 | 0.154 |
| | post-LD | -1.71 | 0.002 | -0.14 | 0.841 |
| OM | LD | -9.51 | 0.000 | +2.35 | 0.423 |
| | post-LD | -17.55 | 0.000 | +2.97 | 0.326 |
| EC | LD | -1.53 | 0.000 | +0.79 | 0.276 |
| | post-LD | -2.27 | 0.000 | +2.07 | 0.006 |
| MD | LD | +1.89 | 0.187 | +6.96 | 0.003 |
| | post-LD | +0.51 | 0.726 | +11.55 | 0.000 |

Pre-LD, LD, and post-LD represent pre-lockdown, lockdown, and post-lockdown period, respectively. "-" and "+" represent "decrease" and "increase", respectively.

9. L236-238: The percentage of SNA decreased during and post lockdown, but the reason behind this behaviour is not discussed.

**Response**: We revised it accordingly.

L274-287: *"Compared with the pre-LD period, the concentration of sulfate (p < 0.01), nitrate (p < 0.01), ammonium (p < 0.01), OM (p < 0.001), and EC (p < 0.001) decreased substantially due to the decline in the emission intensity under the strict control measures during the LD period (Figure 4a, Table S10). The percentage of sulfate (not significant for LD and p < 0.01 for post-LD), nitrate (not significant for LD and p < 0.05 for post-LD), and ammonium (p < 0.05 for LD and p < 0.01 for post-LD) decreased continuously during LD and post-LD, while the MD (p < 0.01 for LD and p < 0.001 for post-LD), OM (not significant for both two periods), and EC (not significant for LD and p < 0.01 for post-LD) increased (Table S10). The mean value of RH declined continuously from pre-LD to LD and post-LD, while the mean value of WS showed an opposite trend (Figure S11). The lower RH and higher WS were not conducive to the secondary formation and accumulation of SNA. Therefore, due to the emission reduction and improved atmospheric conditions, the proportion of SNA decreased sufficiently (from 55.3% in pre-LD to 40.1% in LD and 21.6% in post-LD)."*

10. Since the main objective of the study was identifying changes in emissions during the lockdown, why PMF was not applied to conduct an additional analysis of sources prior and during lockdown?

**Response**: We conducted a study on the impact of COVID-19 lockdown measures on $PM_{2.5}$ sources, and revised the abstract, discussion, and conclusion part accordingly.

11. How did the apportionment to $PM_{2.5}$ change during the lockdown? This is not reported.

**Response**: We added some discussions to section 3.3 and revised the abstract and conclusion section.

L383-399: *"During the LD period, the contribution of SIA, CC, CS, BB, CD, and VE was 22.6, 18.2, 7.7, 5.6, 3.0, and 2.6 µg/m³ to $PM_{2.5}$, respectively, accounting for 37.8%, 30.5%, 12.9%, 9.4%, 5.1%, and 4.4% of the total $PM_{2.5}$ mass (Figure 7). The contribution of CC and dust source (the sum of CS and CD) during the LD period in Hohhot was much higher than those of Tangshan (Wang et al., 2021), Taiyuan (Wang et al., 2022), and Xiamen (Hong et al., 2021) (Table S11). The contribution of SIA was lower than Tangshan and Taiyuan, while higher than Xiamen. Hohhot, Tangshan, and Taiyuan are located in northern China, and consume large amount of coal for winter heating. The high intensity of gaseous precursors emitted from coal combustion is reasonable for a high contribution of SIA. The contribution of VE in Hohhot was lower than Xiamen and Taiyuan. The contribution of VE decreased from 35.5% to 4.4%, whereas the SIA increased from 21.1 % to 37.8 %. The substantial reduction*

*in VE was associated with the strict traffic restrictions during the LD period, which is consistent with the findings in Taiyuan (Wang et al., 2022). Compared with the LD period, the contribution of VE increased from 4.4% to 14.7% during the post-LD period, which can be ascribed to the canceled traffic restrictions. The contribution of CC increased from 30.5% during the LD period to 68.7% during the post-LD period, while the concentration decreased from 29.2 to 18.2 μg/m³. The contribution of SIA decreased from 37.8% during the LD period to 5.0% during the post-LD period, which can be attributed to the improved atmospheric conditions."*

L29-30: *"The contribution of secondary inorganic aerosols increased (from 21.1 to 37.8%), whereas the contribution of vehicular emissions was reduced (35.5% to 4.4%) due to lockdown measures."*

L417-421: *"The source contribution of secondary inorganic aerosols and vehicular emission decreased during the lockdown period, whereas coal combustion increased. The substantial reduction in the contribution of vehicular emissions was associated with the strict traffic restrictions during the lockdown period, the increase in vehicular emission contributions during the post-lockdown period can be attributed to the canceled traffic restrictions."*

[Figure]

***Figure 7.*** *(a) Concentration and (b) percentage of source contribution to PM$_{2.5}$ in Hohhot in spring, summer, autumn, winter, over the sampling year, pre-lockdown, lockdown, and post-lockdown. CC, VE, SIA, CD, CS, and BB represent coal combustion, vehicular emission, secondary inorganic aerosol, construction dust, crustal sources, and biomass burning, respectively.*

12. Overall, the text is clear and understandable but there are some sentences that require re-writing and re-wording (L95-inhabitants?, L114-analysis, L181, L222-contributed by X % to total $PM_{2.5}$..., 278-benefical?, L283-easier, you meant faster?... ).

**Response**: We checked through our manuscript and corrected the typos accordingly. (L101-102, L121, L254, L262, L267, L327, and L335-336)

---

## Author Response (AR2)

**Responses to editor**

We would like to thank the editor for the helpful comments on our manuscript. We have studied the comments carefully and revised our manuscript accordingly, which can be found in the attached file (Track Changes). Our point-by-point replies to the comments are provided below. Referee comments are given in black, and our replies are given in blue.

**Comments to the author:**

I would like to thank the authors for incorporating the suggestions made by both reviewers. The revised version looks pretty good and it is almost ready for publication; however, I have the following additional comments before I can accept the manuscript.

Minor/Technical Comments:

Line 25: I think it should be "declined"

      **Response**: We revised it.

Line 28: (from 35.5% to 4.4%)

      **Response**: We revised it.

Line 34: Add a reference after (2.5 um)

      **Response**: We revised it.

Line 36: Add an space between "2015" and "(MEEC"

      **Response**: We revised it.

Line 39: Add a reference after "heating"

      **Response**: We revised it.

Line 44: Add a reference after "PM2.5"

>**Response**: We revised it.

Line 57: Add a reference after "measures"

>**Response**: We revised it.

Line 59: Add a reference after "reduction"

>**Response**: We revised it.

Line 65: Add a reference after "winter"

>**Response**: We revised it.

Line 75: Add a reference after "period"

>**Response**: We revised it.

Lines 76-77: This was somehow already mentioned in lines 65-66

>**Response**: We deleted it.

Line 79: during the LD period. It should be "." instead of ","

>**Response**: We revised it.

Line 97:..part of THE Inner...

>**Response**: We revised it.

Line 100:...with THE Daqing...

>**Response**: We revised it.

Line 100:...and THE Manhan...

>**Response**: We revised it.

Line 102: Add a reference after "summer"

**Response**: We revised it.

Line 103: Add a reference after "spring"

**Response**: We revised it.

Line 104: Add a reference after "year"

**Response**: We revised it.

Line 106: Please add at how many meters above the ground the samplers were placed and also briefly described the surroundings of the sampling site.

**Response**: We revised it.

***Revised text:*** *The sampling site was located on the rooftop (approximately 50 m above the ground) of the main building of the Ecological and Environmental Department of the Inner Mongolia autonomous region (Figure 1) and represents a typical semi-arid urban environment. The sampling site is surrounded by residential areas without industrial sources nearby. There is one main road named Tengfei Road 50 m to the east.*

Line 118: ...stored at -18C. In sealed Petri dishes?

**Response**: Every filter was stored in a preservation box (the following picture) separately.

[Figure]

***Revised text:*** *After weighing, every filter was stored in a preservation box separately at -18 ℃ until analysis.*

Line 119: Please add the used SO2, NO2, CO, and O3 sensors

**Response**: We revised it.

**Revised text:** *The hourly concentrations of $SO_2$, $NO_2$, CO, and $O_3$ were measured using T100, T200U-NOy, T300, and T400 sensors produced by Automated Precision Inc. (USA), respectively.*

Line 124: WSIs was already defined in line 113

**Response**: We revised it.

Line 125 and 126: OC and EC were already defined in line 113. Used it here

**Response**: We revised it.

Line 138:...using equations 1-2

**Response**: We revised it.

Line 138: Add a reference after "1-2"

**Response**: We revised it..

Lines 140-141:...using equations 3-5...

**Response**: We revised it..

Line 154: Define "Q/Qexpected"

**Response**: It is an output parameter of PMF. It can not be defined. We revised the sentence and added some references.

**Revised text:** *According to the changes in an output parameter ($Q/Q_{expected}$) and estimation diagnostics analysis (Ulbrich et al., 2009; Tian et al., 2020; Liu et al., 2021), six factors solutions were selected.*

Line 158: Here "profiles of PM2.5" is mentioned; however, in figures S1-S2 PMF is

mentioned instead of PM2.5. Please be consistent

**Response**: We revised it.

Line 162: I suggest moving here lines 200-205.

**Response**: We revised it.

Line 162: I am wondering what the authors mean with "dynamically"

**Response**: We deleted it.

Line 170: How about panel 2e?

**Response**: We revised it.

Line 185: Add a reference after "period"

**Response**: It is a conclusion of our result. The conclusion is based on the following reasons: "Firstly, chloride in heating period is significantly higher than that in non-heating period. Secondly, Hohhot is an inland city, basically unaffected by sea salt. Thirdly, a higher average $Cl^-/Na^+$ ratio (3.43 for January) suggests the presence of non-marine anthropogenic sources of chloride." We revised the sentence to make it more clear.

Line 191: Add a reference after "RH"

**Response**: We revised it.

Line 201: Add a reference after "2020"

**Response**: We revised it.

Line 212: Change "result" with "results"

**Response**: We revised it.

Line 212: Add a space between "2015" and "(Wang"

**Response**: We revised it.

Line 217: Change "indicates" with "suggests"

**Response**: We revised it.

Line 222: Replace "relative humidity" with "RH"

**Response**: We revised it.

Line 222: Replace "wind speed" with "WS"

**Response**: We revised it.

Line 224: Add a reference after "area"

**Response**: We revised it.

Line 240: "(25th December, 2019 to 24th 240 January, 2020)". This was already defined

**Response**: We deleted it.

Line 256: ...ranked as pre-LD:.............; LD:....and; post-LD:.....NH4+ (1.8%). Compared...

**Response**: We revised it.

Lines 276-278: Please confirm that the "Less than" and "Less than or equal to" symbols are correctly used/placed.

**Response**: We revised it.

*Revised text: Figure 5.* *(a) Concentrations and (b) percentages of chemical components in $PM_{2.5}$, (c) meteorological conditions, SOR, NOR, and OC/EC at different pollution levels and during different types of pollution events. CP, SP, MP, and HP represent the clean ($PM_{2.5} < 35$ $\mu g/m^3$), slightly polluted ($35 \leq PM_{2.5} < 75$ $\mu g/m^3$), moderately polluted ($75 \leq PM_{2.5} < 150$ $\mu g/m^3$), and heavily polluted periods ($PM_{2.5} \geq 150$ $\mu g/m^3$ ), respectively.*

Line 287: Add a ")" after "m3"

**Response**: We revised it.

Line 290: Add "%" after "2.4"

**Response**: We revised it.

Line 318: Replace "was" with "were"

**Response**: We revised it.

Line 338: Define "BB"

**Response**: We revised it.

Line 397: There are two dots. Please delete one of them.

**Response**: We revised it.

**References**

Liu, Y., Li, C., Zhang, C., Liu, X., Qu, Y., An, J., Ma, D., Feng, M., and Tan, Q.: Chemical characteristics, source apportionment, and regional contribution of PM2.5 in Zhangjiakou, Northern China: A multiple sampling sites observation and modeling perspective, Environ. Adv., 3, 100034, https://doi.org/10.1016/j.envadv.2021.100034, 2021.

Tian, Y., Zhang, Y., Liang, Y., Niu, Z., Xue, Q., and Feng, Y.: $PM_{2.5}$ source apportionment during severe haze episodes in a Chinese megacity based on a 5-month period by using hourly species measurements: Explore how to better conduct PMF during haze episodes, Atmos. Environ., 224, 117364, https://doi.org/10.1016/j.atmosenv.2020.117364, 2020.

Ulbrich, I. M., Canagaratna, M. R., Zhang, Q., Worsnop, D. R., and Jimenez, J. L.: Interpretation of organic components from Positive Matrix Factorization of aerosol mass spectrometric data, Atmos. Chem. Phys., 9, 2891-2918, https://doi.org/10.5194/acp-9-2891-2009, 2009.